# Research Progress of Low Density and High Stiffness of Be-Al Alloy Fabricated by Investment Casting

**Junyi Li [1], Yao Xie [1,2], Yiqun Yang [1], Zhaogang Liu [1], Dongxin Wang [1,*] and Yajun Yin [2]**

1  State Key Laboratory of Special Rare Metal Materials, Northwest Rare Metal Materials Research Institute Ningxia Co., Ltd., Shi Zuishan 753000, China
2  State Key Laboratory of Materials Processing and Die & Mould Technology, Huazhong University of Science & Technology, Wuhan 430074, China
*  Correspondence: wangdongxin123@126.com; Tel.: +86-18095210234

**Abstract:** Be-Al alloy is a type of in situ metal matrix composite composed of a primary Be phase for strength and stiffness and a continuous Al matrix for ductility and toughness. Be-Al alloy (AlBe-Cast®910) has the characteristics of low density (2.17 g/cm³), high elastic modulus (193 GPa) and specific stiffness (88.94 GPa/(g/cm³)) as a preferred material for lightweight aerospace products. Investment casting technology can be employed to prepare the components with thin-walled complex structures for aerospace; however, the wide solidification range for Be-Al leads to difficulty in feeding a casting and results in extensive shrinkage and porosity in cast parts. In this paper, the characteristics of Be-Al alloy are introduced first. Secondly, the mechanisms of influence of adding elements on the casting process, mechanical properties (strength increases more than 20% by adding elements) and microstructure evolution are explained in detail. In addition, the heat treatment technology (strength increases at least 10% after heat treatment) and the repair of defects by electron beam welding are discussed. Finally, Be-Al alloy is a new type of composite material, and China is a major research and application country; this paper introduces its research status and analyzes existing problems and shortcomings and points out the direction of Be-Al alloy development in China in the next few years.

**Keywords:** Be-Al alloy; high elastic modulus; adding elements; mechanical properties; microstructure evolution

## 1. Introduction

Be-Al alloy combining Be with Al takes the advantages of the excellent ductility of the Al phase and the high modulus of Be phase [1,2]. The alloy has the characteristics of lightweight, high specific strength, high specific stiffness, good thermal stability, high toughness, high modulus, and corrosion resistance. As the solid solubility between Be and Al is very low, Be-Al alloys with properties substantially different from those of pure metals should be defined as a composite material when the Be content is below 65 wt.%, in which discontinuous granular Be phase reinforces continuous Al phase as a matrix material [2]. Be-Al alloy has been widely used in missile, airborne and spaceborne platforms in Europe and the United States [3]; moreover, it has been listed as a key material in the development of kinetic energy interceptors by the United States [4]. Be-Al alloy was first developed in the 1960s by the United States. Nowadays, the alloy is becoming the key material for high-end equipment, such as kinetic energy interceptors, airborne situational awareness devices, and directed energy weapons [5,6].

Be-Al alloy is an important feature of military applications, and its research status and development trend are of great concern to everyone; this paper studies and analyzes the research development status and existing problems of Be-Al alloy prepared by investment casting method in China, which is of great significance to promote the development of this material in China.

The employees who are exposed to Be (dust) could lead to the more-common chronic Be disease and/or to Be sensitization during the manufacturing of Be alloy [7]. Be-Al alloy is a typical alloy with a high beryllium content. Consequently, toxicity protection of Be should be taken into account in the research and production process; however, it should be mentioned that the toxicity of Be and Be-Al alloy has no negative impact on the use of the product.

The investment casting process is commonly employed to prepare complex components, and it has the characteristics of less processing, which can save a lot of processing time and cost and also improve the utilization rate of materials. Therefore, the investment casting process is one of the main preparation methods for Be-Al alloy. The casting technology developed for the Al alloy and Ti alloy is applied to the near net shape of Be-Al alloy. More specifically, the casting process meets the standard cast Al design guidelines and requires minimal design changes. The same basic equipment for Al investment casting is used in the casting process of Be-Al alloy, and the only unique features of the Be-AlCast process are the need for particulate collecting equipment and the use of vacuum casting versus the more traditional air melt casting process [4]; however, the absence of compounds and low mutual solubility both hinder the production of Be-Al alloy with beneficial properties substantially. The above problem can be mitigated via alloying-induced microstructure modification and mechanical reinforcement.

Investment casting and additive manufacturing (AM) are both near-net forming technologies. AM is a very promising technology permitting the development of three-dimensional (3D) printed models or functional parts with precise and complex geometries such as those obtained from a topology optimization process [8,9]. Electron beam powder bed fusion (EB-PBF), laser beam powder bed fusion (LB-PBF), fused filament fabrication (FFF), material jetting, direct energy deposition and material extrusion are common AM methods to produce metallic components [10]. Ti6Al4V [11], IN718 [12], 316L [13], 316Lwith ARCH lattice structure [14] and Al6061 [15] alloys are the most common case studies. The commercial Be-Al products are not currently produced by AM processes directly [16] but are being produced indirectly by AM-generated patterns (SLS) used for investment casting [17].

Many elements have been added to improve the casting performance and reduce casting defects. As a result, the advantages of investment cast Be-Al alloys over other lightweight alloys are obvious, as shown in Table 1. The AlBeCast 910 composite is a type of ternary Be-Al-Ni composite; it has a relatively high modulus and high specific modulus, importantly. As a result, the structural stability of Be-Al alloy is excellent.

**Table 1.** The properties of Be-Al alloy and other light alloys [18].

| Property | AlBeCast®910 | Aluminum A357-T6 | Titanium 6AL-4V | AlSiC F3A205-T6 |
|---|---|---|---|---|
| Density g/cm$^3$ | 2.17 | 2.69 | 4.43 | 2.69 |
| CTE (25 °C) ppm/°C | 14.6 | 21.5 | 9.3 | 16.4 |
| Modulus GPa | 193 | 72 | 110 | 72 |
| Yield Strength, MPa | 165 | 248 | 880 | 165 |
| Ultimate Tensile Strength, MPa | 211 | 317 | 950 | 196 |
| Specific Stiffness, GPa/(g/cm$^3$) | 88.94 | 26.77 | 24.83 | 26.77 |

This paper introduces and summarizes the investment casting process and heat treatment of Be-Al alloy, and it may be served as reference significance for the development of Be-Al alloy.

## 2. The Characteristics of Investment Be-Al Alloy

Be-Al alloy is a combination of high stiffness and low density of Be element with easy processability characteristics of Al element. The binary alloy phase diagram is illustrated in Figure 1a, and the temperature of the eutectic reaction is about 644 °C. The atomic fraction of Be element at the eutectic point is only 2.4 at.%, and there are no intermetallic compounds. Subsequently, the separation of Be and Al elements occurs in the process of solidification.

Based on the properties of the Be-Al binary system, the two phases can be separated without nucleation; finally, the special three-dimensional network structure [19] of Be phase (62 wt.% Be) and Al phase (38 wt.% Al) is shown in Figure 1b from the X radia Context Micro CT will be formed, in which, Be embedded in the Al matrix in columnar crystal form. The solid solubility of Be in the Al phase is 0.1 wt.%, and the solid solubility of Al in Be phase is only 0.02 wt.% [20]. Therefore, Be-Al alloy should be defined as a composite material when the atomic fraction of Be is in the range of 60–80 at.%. The discontinuous granular Be phase reinforces the matrix of the continuous Al phase shown in Figure 1b. In the scanning electron microscopy (SEM) image of the Be-Al alloy, the dark imaging phase in the microstructure is the Be phase (62 wt.% Be), and the grey imaging matrix is the Al phase (38 wt.% Al), as shown in Figure 2. The casting difficulty of Be-Al alloy is significantly greater than other common alloys due to the particularity of the solidification process. The growth of pure primary Be dendrites is within the Al matrix during solidification and thus hinders the production of Be-Al alloys with beneficial properties, which are substantially are close to those of pure constituents [21].

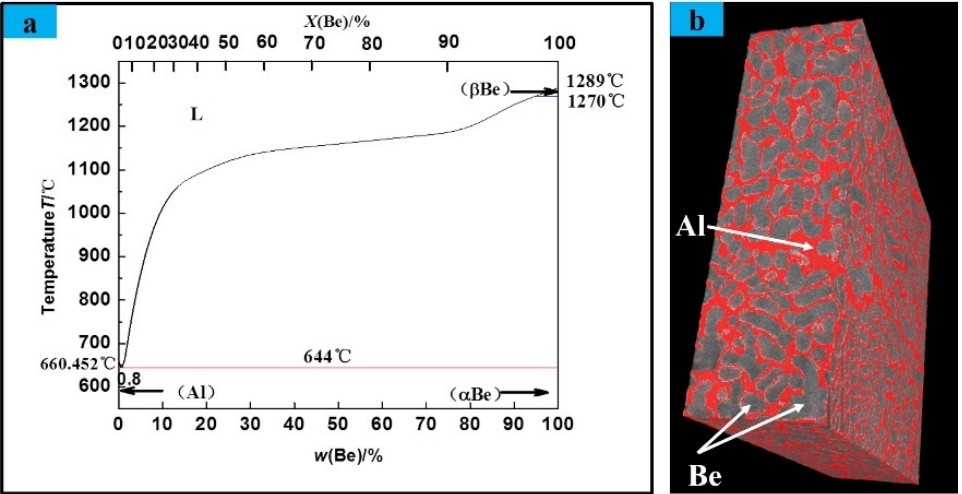

**Figure 1.** The phase diagram of Be-Al alloy (**a**), three-dimensional microstructure (**b**) [19]. Reprinted/adapted with permission from Ref. [19]. Copyright 1994, Elsevier Ltd.

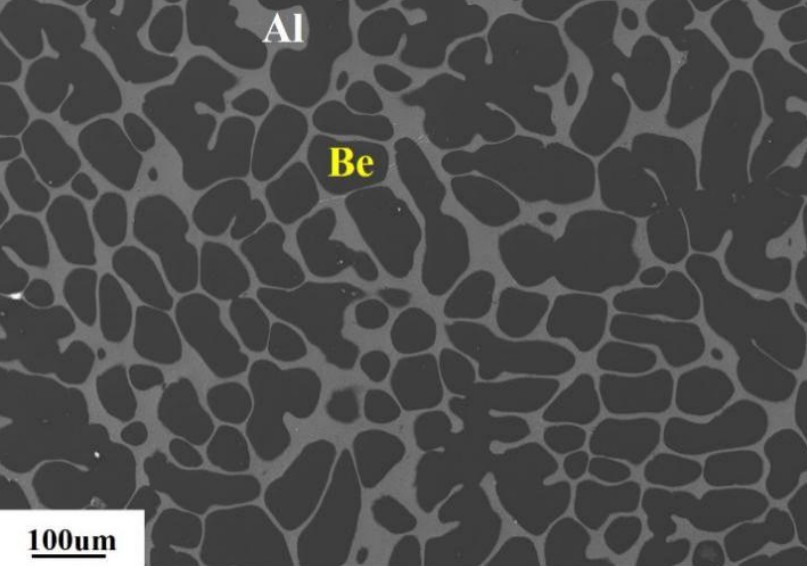

**Figure 2.** The SEM image of Be-Al alloy.

The surface energy of a binary Be-Al alloy can be expressed from a regular solution model using mean-field approximation statistics [22].

$$\lambda = ZN_a[\varepsilon_{BeAl} - (\varepsilon_{BeBe} + \varepsilon_{AlAl})/2] \tag{1}$$

Z is the mean coordination number in the liquid, $N_a$ is the Avogadro's number, and $\varepsilon_{BeAl}$ is the interaction energy of a Be-Al pair. $\lambda$ (the exchange energy) from Equation (1) is 25 KJ/mol [23], The heats of evaporation for Be ($-ZN_a\varepsilon_{BeBe}/2$) and Al ($-ZN_a\varepsilon_{AlAl}/2$) [24] are 297 KJ/mol and 294 KJ/mol, respectively. Thus, the value of heats of evaporation for Be-Al ($-ZN_a\varepsilon_{BeAl}/2$) is only 283 KJ/mol calculated from Equation (1), which is lower than the data of Be and Al, indicating that strong interactions between Be and Al cannot happen.

On the other hand, Figure 3 showed bright-field TEM micrographs of the phase interface between Be and Al. Be(0001) and Al(110) formed a non-coherent interface, and there are no interfacial intermediates and transitional phases.

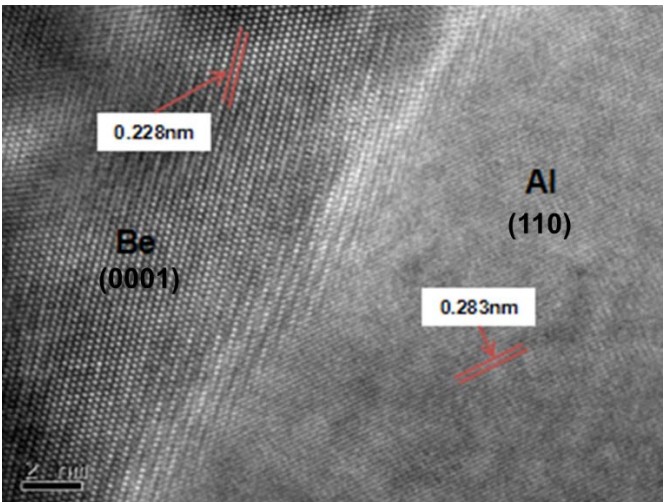

**Figure 3.** Bright field HRTEM images of the phase interface between Be and Al.

### 3. The Effect of Adding Elements on the Casting Process

Some elements are often added to improve the mechanical properties and decrease casting defects of Be-Al alloy [25], such as Ag [26,27], Si, Cu, Ni, Co [26], and Mg [28]. Allowing for the basic principles of alloy strengthening, the solubility of elements in aluminum and beryllium is firstly considered. The elements would be selected which could form a noticeable solid solution with both Al and Be.

On the basis of analysis of the solubility of the adding elements in Al and Be. These elements are used for alloying in Al–X systems (X = Ag, Cr, Cu, Hf, Li, Mg, Mn, Sc, Si, Ti, V, Zn, Zr). Components X = Ag, Au, Cu, Fe, Co, Ni, and Pd are likely to have a noticeable solubility in Be.

The properties of Be-Al alloy are listed in Table 2 with different adding elements. Whether in China or the United States, Be-Al (Ag/Co/Ge) with the best properties is put to wide use in the commercial application mainly. Be-Al alloy (Ag/Co/Ge) has been applied widely, such as AlBeCast 920® in Materion Inc.; moreover Bery1Cast363® in Starmet Inc.

The Be-Al alloy (62Be-33.25Al-3Ag-1Co-0.75Ge) was based on "Lockalloy" (62Be-38Al by weight). High-purity Al ingots (99.99%), vacuum-purified Be ingots (99.9%), 3 wt.% Ag (99.99%), 1 wt.% Co (99.99%), and 0.75 wt.% Ge (99.99%) were used as the raw materials. All the ingots were abraded to remove superficial oxides, cleaned by sonication in ethanol, and dried in nitrogen atmosphere at 80 °C for over 1 h prior to vacuum induction melting. The raw materials were melted and homogenized at 1400 ± 10 °C for 20 min before being poured into the ceramic shell which was heated at (600 ± 10) °C before. In the end,

argon gas was applied as a cooling medium to accelerate the solidification process under 40–70 °C/s measured from an infrared thermometer in the cooling chambers.

**Table 2.** The properties of Be-Al alloy with different adding elements [29–31].

| The Samples | Tensile Strength MPa | Yield Strength MPa | Elongation (%) |
|---|---|---|---|
| 62Be-38Al alloy | 110.5 | 91.7 | – |
| [a] 65Be-30.25Al-3Ag-1Co-0.75Ge | 284.0 | 214.0 | 4.8 |
| [b] 62Be-33.25Al-3Ag-1Co-0.75Ge | 243.0 | 185.0 | 2.5 |
| 62Be-37.6Al -0.4Sc | 179.7 | 110.7 | 2.9 |
| 65Be-31Al -2Si-2Ag | 196.0 | 137.0 | 1.7 |
| 62Be-34Al -4Ni | 211.0 | 165.0 | 4.0 |

Note: [a] 65 wt.%Be-Al (Ag/Co/Ge) from Materion inc. in the United States. [b] 62 wt.% Be-Al (Ag/Co/Ge) from Northwest Rare Metal Materials Research Institute Ningxia Co., Ltd. in China.

The effect of adding elements on the casting process listed in the Table 3 to summarize the main findings from Section 3.

**Table 3.** The effect of adding elements on the casting process of Be-Al alloy.

| Element | Microstructural Evolution | Advantages Due to the Added Element | Reference |
|---|---|---|---|
| Ag/Co/Ge | 1. Ag, Ge are solidly soluble in Al 2. $Ag_2Al$ or other Ag-Al binary alloy formed | 1. Ag improve the casting performance, strength, and plasticity of the alloy 2. Ge is mainly to improve the fluidity, strength, and toughness of the alloy 3. Co increases the strength and hardness of the Be phase, then improves the strength and hardness of the alloy | Reference: [26,27,29,31–35] |
| Ni | $\alpha$Be (Ni is solidly soluble in Be) | reduces the thermal expansion coefficient, increases tensile properties and improves castability | [18,33,36,37] |
| Sc/Zr | second-phase particle $Al_3Sc$, $Be_{13}Sc$ and $Be_{13}Zr$ formed | 1. Be grain refinement 2. the secondary dendritic arm spacing reduced | [30,38–47] |
| Si | 1. irregularly shaped particle within the Al matrix 2. precipitate in the alloy | 1. enhances the strengthening effect of Be-Al alloy | [31,48] |
| Cu | 1. $\alpha$Be (Cu is solidly soluble in Be) 2. $\alpha$Al (Cu is solidly soluble in Al) | precipitate and harden the Al phase | [49–51] |

Ag, Co, Ge, Ni, Sc, Si and Cu elements are introduced in this paper to improve its casting properties and mechanical properties, mainly by strengthening the Al phase defined as the matrix material to improve the alloy strength.

### 3.1. Ag, Co, and Ge Elements

Ag, Co, and Ge are the main adding elements selected to improve the mechanical properties of the Be-Al alloy.

The distribution of different elements was analyzed respectively by surface scanning energy spectrum in Figure 4. The Be-Al alloy is 62 wt.% Be-33.25 wt.% Al (3 wt.%Ag/ 1 wt.% Co/0.75 wt.% Ge). The Be as the reinforced phase is columnar dendritic as shown in Figure 4a, and the Al is a continuous phase as shown in Figure 4b. The distribution of Ag (Figure 4c) and Ge (Figure 4e) is consistent with the Al; it is confirmed that the Ag and Ge are solidly soluble in the Al phase. The Co as shown in Figure 4d is solidly soluble in the Be phase.

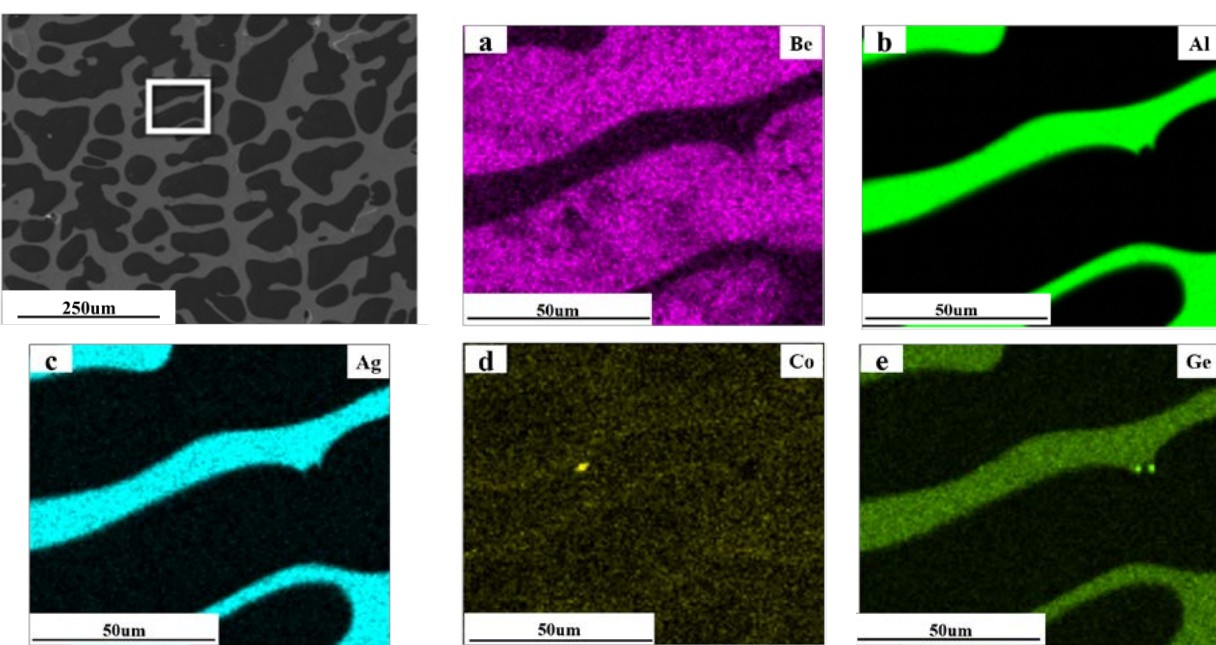

**Figure 4.** Mappings of different elements in Be-Al alloy (**a**) Be, (**b**) Al, (**c**) Ag, (**d**) Co and (**e**) Ge [32].

Some research shows that the microstructure of Be-Al alloy is significantly changed after the addition of Ag element. Be particles and about 40% (atomic fraction) δ phase Al are dispersed in Al-Ag matrix. The δ phase of Al is lower than the α phase in terms of strength and solid solubility of Ag atom. In general, it is necessary to heat the alloy at 500~570 °C for 1~2 h, then the δ phase of Al can be completely transformed into α phase, so that Ag can be completely dissolved in Al [27]. In first-principles study [33], it is investigated that the overlapping population of Be and Al atoms around the interface increases with Ag addition, which makes the bonding between Be and Al atoms at the interface stronger. Therefore, it is beneficial to improve the interface bonding/plasticity of Be-Al alloy. Furthermore, the research indicated that Ag only exists in the Al phase in Be-Al alloy, which can improve the casting performance, strength, and plasticity of the alloy [27]. Figure 5 shows the distribution of the Ag rich precipitation strengthening phase in the Al matrix. Precipitation hardening is the large region of solid solubility of Ag in Al near the eutectic temperature of Ag-Al system. All of Ag will be in solid solution an Al alloy with the content of Ag up to 23 at.% at an elevated temperature. Subsequently, upon quenching, the Ag can be trapped in a supersaturated solid solution, which can then decompose in ways that can result in additional hardening of the alloy.

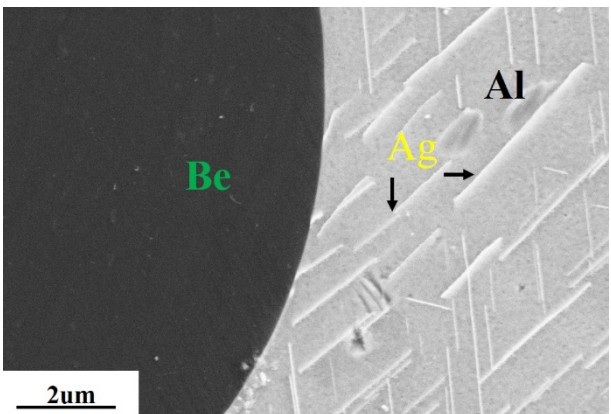

**Figure 5.** The SEM images of Ag precipitated phase.

The effect of Ge is mainly to improve the fluidity, strength, and toughness of the alloy. Compared with the addition of Si, It is more effective to reduce the microporosity and increase the strength and ductility of Be-Al alloy by the addition of Ge element [26].

Both Co and Be are close-packed hexagonal crystal structures, as a result, Co can be solidly soluble in the Be phase. The addition of Co increases the strength and hardness of the Be phase, then improves the strength and hardness of the alloy [34]; it is easy to combine with Be atom and Al atom, then to form a metal bond, which reduces the number of covalent bonds at alloy phase interface and improves the toughness of alloy phase interface [33]; moreover, the addition of Co is beneficial for the nucleation of Be phase to reduce the coarse dendrite and columnar crystal structure, and refine Be grain and improve the strength of Be-Al alloy [35].

### 3.2. Ni Element

Be-Al alloy containing Ni is used in fields requiring high plasticity. The additional proportion of the Ni element is usually about 4%. The microstructure and fracture mechanism of Be-Al alloy did not change with the addition of Ni, while the tensile strength and yield strength of the alloy with 4 wt.% Ni are 234.1 MPa and 158.9 MPa, respectively [36]. The purpose of solution strengthening has been achieved as the distribution of Ni element is mainly in the Be region without forming the second phase, as shown in Figure 6. According to research results, Ni could migrate and disperse in Be phase when 3 wt.% Ni was added into Be-Al alloy, and the interface bonding strength of Be-Al alloy can be enhanced [37]; it was obtained from first-principles calculations that addition of Ni had a positive effect on reducing the brittleness of alloys, but it has no effect on the interface bonding strength of Be-Al alloy [33].

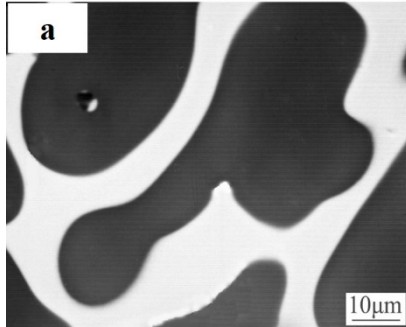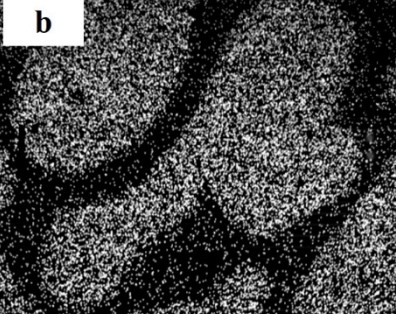

**Figure 6.** The microstructure of Be-Al alloy (**a**) and Ni element distribution (**b**).

### 3.3. Sc Element

Sc is one of the most effective alloying elements for Al alloys due to exhibiting the largest matrix strengthening effect and increasing the recrystallization temperature; moreover, Sc is also the strongest modifier of the as-cast grain structure due to inducing significant grain refinement [38] with a content below 0.6 wt.%, which is ascribed to the formation of primary $Al_3Sc$ particles during crystallization and ease in the homogeneous origin of secondary phase particles in the decomposition of the supersaturated solid solution of Sc in Al [39]. In addition, Sc exhibits limited solubility in Be and forms a series of intermetallic compounds [40]. The Al-Be-Sc ternary phase diagram was studied by Raghavan [41,42]. The ternary eutectic temperature is 1280 °C (0.1~0.15%, atomic fraction) on the Be-rich side of the alloy. Crystallization in the ternary phase depends on the behavior of $Al_3Sc$ and $ScBe_{13}$ compounds.

The microstructure and mechanical properties of Be-Al alloy with 0.2~3 wt.% Sc were investigated. The primary principal axis, secondary dendrite arm, and dendrite spacing of Be in the alloy are effectively reduced by 0.2~3 wt.% Sc adding, meanwhile, there is the best when adding 0.4 wt.% Sc. The $Be_{13}Sc$ and $Al_3Sc$ phases in the alloy will be formed with the addition of Sc element, but both the phases did not act as effective nuclear particles

of Be matrix. Sc-Zr alloy can reduce the secondary dendrite arm spacing (SDAS) of Be-Al alloy compared with a single Sc element [43,44]. Figure 6 shows the microstructure of as-cast Be-Al, Be-Al-Sc, and Be-Al-Sc-Zr alloys prepared using the VAM technique, with the two distinctly separated Be and Al phases marked by red and blue arrows, respectively. Whereas the Be–Al alloy exhibited a pronounced coarse columnar dendritic microstructure (Figure 7a), the addition of Sc decreased SDAS of the Be phase by approximately 1/3 (from 19 to 12 μm) and resulted in Be grain size refinement, and the obtained cellular/equiaxed microstructure depicted in Figure 7b. What is more, the SDAS of Be decreased to ∼7.5 μm in Be–Al–Sc–Zr, with the morphology of the Be phase there being similar to that of the equiaxed microstructure, featuring further refined Be grains (Figure 7c) [45].

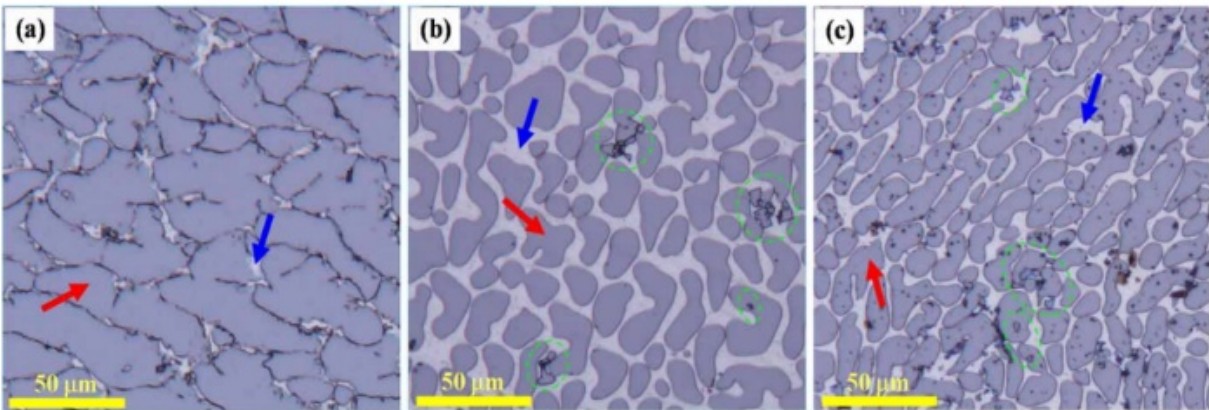

**Figure 7.** The OM images of as-cast alloys:(**a**) Be-Al, (**b**) Be-Al-Sc, and (**c**) Be-Al-Sc-Zr alloys [45] Reprinted/adapted with permission from Ref. [45]. Copyright 2018, Taylor & Francis Ltd.

The microstructural and compositional characterisations were investigated by SEM and EDS in Figure 8. The dark and bright areas in SEM micrographs were ascribed to Be and Al phases, respectively. The EDS analysis was performed within the highly faceted phase regions marked by green dashed-line circles.

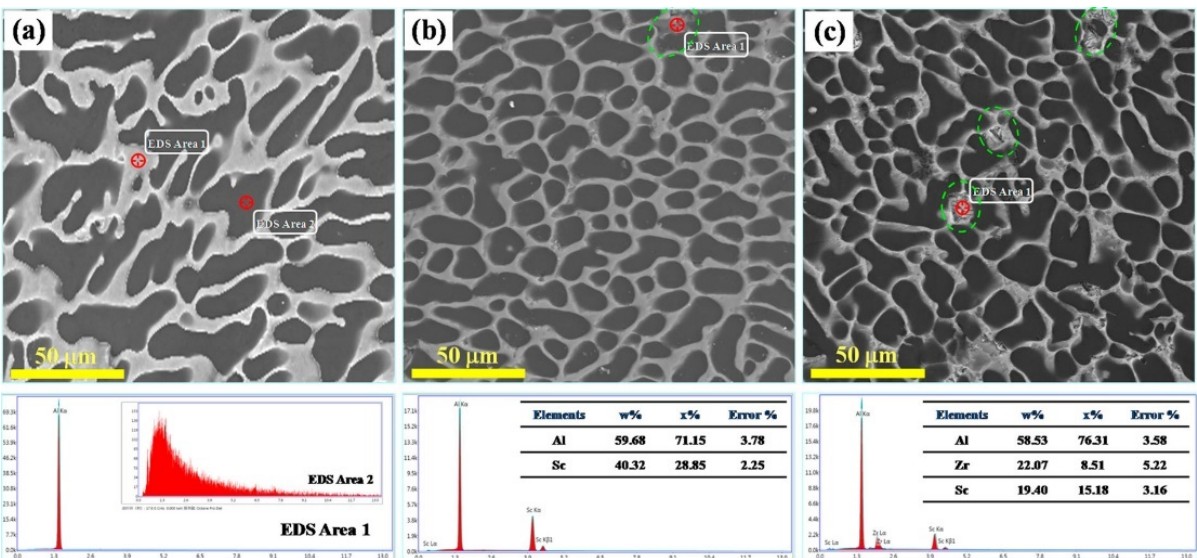

**Figure 8.** SEM micrographs of as-cast (**a**) Be–Al, (**b**) Be–Al–Sc, and (**c**) Be–Al–Sc–Zr alloys with corresponding area-scanning EDS profiles of marked highly faceted phases [45]. Reprinted/adapted with permission from Ref. [45]. Copyright 2018, Taylor & Francis Ltd.

Only Al was detected in the Al phase of the Be–Al alloy, and Be could not be detected by EDS directly may caused by the very low fluorescence yield of Be and the strong absorption of Be radiation in the sample [46] whereas Sc and Sc–Zr-alloyed specimens comprised significant amounts of Sc and Zr. The EDS and XRD characterization of Be–Al–Sc–Zr indicated that its highly faceted phases contained both $Be_{13}Sc$ and $Be_{13}Zr$. Notably, the existence of $Al_3Sc$ and $Al_3(Sc_{1-x}Zr_x)$ had to be revealed via further characterization.

Figure 9 shows a representative TEM image of as-cast Be–Al–Sc–Zr, EDS characterization of both phases revealed the presence of Al, Sc, and Zr within phase A, whereas only Sc and Zr were detected within phase B; moreover, selected-area diffraction (SAD) patterns of the two phases were collected, with that of phase A resembling that of previously reported $Al_3Sc$ particles [47]; however, the lattice parameters of phases A and B were calculated as 0.406 nm and 1.009 nm, respectively. The substitution of one part of Sc lattice positions by Zr atoms in the crystal, which resulted in lattice parameter decrease [38], as a results, the distinct variance between the experimental and reported lattice parameters of $Al_3Sc$ appeared.

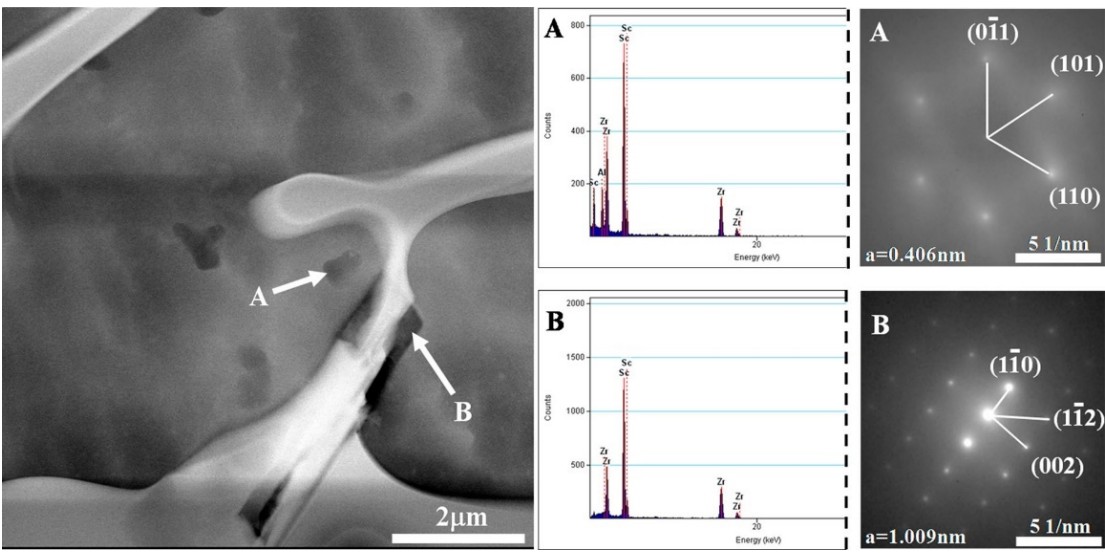

**Figure 9.** TEM image of the as-cast Be–Al–Sc–Zr alloy with corresponding EDS and SAD patterns of phases (**A**,**B**) [45]. Reprinted/adapted with permission from Ref. [45]. Copyright 2018, Taylor & Francis Ltd.

*3.4. Si Element*

The Be-Al alloy with Si element was developed by Nuclear Metals, inc., in the United States [31,48]. The wettability of each component phase is increased by the Si element and the strength and porosity of the alloy can be improved by adding 0.2–4.25 wt.% Si element. The strength of the alloy can be further improved by adding 0.1–2 wt.% Cu, Ni, and Co. 0.005–0.2 wt.% Sr or Sb can also be added to improve the ductility of the Be-Al-Si alloys.

The Si-rich phase forms as a discreet irregularly shaped particle within the Al matrix phase. The Si particles produce some strengthening of the Al phase. The presence of Si in the Al phase also enhances the strengthening effect of the Al-Ag phase in the alloy [31]. Si element can also precipitate in the alloy structure and accumulate at the grain boundary of the alloy; however, it is easy to be oxidized to form silicon oxide, which causes the brittleness of the grain boundary if the content of Si added exceeds 5%. The experiments indicate that the optimal ratio of Si element is 3% [48]. The 0.005~0.200% Sr or Sb is added into the Be-Al alloy with Si element in order to modify the Si structure and improve the toughness of the ingot; it is found that the yield strength of the alloy is 154.3 MPa, the ultimate tensile strength is 210.8 MPa, the elongation is 2.5%, the density is 2.13 g·cm$^{-3}$, and the elastic modulus is 227.37 GPa when the alloy composition is 65Be-31Al-2Ag-2Si-0.04Sr [31].

*3.5. Cu Element*

It is investigated that Cu and Si can be completely dissolved in the Al phase at 500~550 °C holding for 12 h, which can precipitate and harden the Al phase in the Be-Al alloys (50~85% Be, 13.4~46.5% Al, 0.6~3.5% Cu and 0.3~2.5% Si) prepared by Larsen et al. [49,50]. The melting points of Cu and Be are similar, so it is difficult to maintain the optimal sintering conditions. What is more, the viscosity of the Al-Cu alloy is lower than that of molten Cu because the melting point of Al is far lower than that of Cu. Therefore, Al and Cu are mixed by sintering below the melting point of Be. Accumulation is avoided and complete alloying is hindered in this way, which is favorable for random orientation of most grains and anisotropy of alloy [51].

## 4. The Heat Treatment of Casting Be-Al Alloy

The strength of Be-Al alloy casting can be improved by heat treatment. Both the homogenization and aging treatment are the main heat treatment processes of Be-Al alloy as listed in Table 4.

Because the solution of adding elements in the Be phase is very small and the diffusion rate is low at a certain temperature, the heat treatment process generally refers to the corresponding heat treatment of Al alloy [52,53]. A series of Be-Al alloys(65Be-31Al-2Si-2Ag-0.04Sr) were solution heat treated (HT) for 2 h at 550 °C, and water quenched, then aged for 16 h at 190 °C and air cooled. The mechanical properties are shown in Table 5. Heat treatment caused grain growth and the insoluble phases Si aggregated and coarsened, resulting in an enhancement of tensile strength, similar to Al-Si alloy [54]. The melting temperature of the eutectic Al phase in Al-Si alloy [55] is lower. The process is to enhance the Al phase by Si [56,57] and Ag, mainly.

The temperatures of solution treatment are 475 °C (4.5 h), 505 °C (2.5 h) and 545 °C (4.5 h), separately. Subsequently, the temperatures of artificial ageing are 280 °C (6 h), 165 °C (4 h) and165 °C (6 h), respectively [58].

**Table 4.** The heat treatment of casting Be-Al alloy.

| Heat Treatment | Composition | Advantages Due to the Heat Treatment | Reference |
|---|---|---|---|
| Solution and aging treatment. | 65Be-31Al-2Si-2Ag-0.04 Sr(0.25Cu, Ni and Co) | Strength improved | [31] |
| Homogenizing and aging treatment. | 62Be-37.6Al-0.4Sc | Hardness improved Electrical conductivity improved | [59,60] |

**Table 5.** The mechanical properties of Be-Al alloys (as-cast and heat treated) [31].

| Composition | Conditions | Tensile Strength MPa | Yield Strength MPa | Elongation(%) |
|---|---|---|---|---|
| 60Be-40Al | as-cast | 110.5 | 91.7 | 1.0 |
| 65Be-31Al-2Si-2Ag-0.04Sr | as-cast | 190.3 | 138.6 | 2.3 |
| | heat treated | 217.9 | 158.6 | 2.5 |

The metallographic study was performed to evaluate the microstructure of the investigated EN AC-AlSi11(Fe) alloy, and the process conditions and properties of heat treatment are also shown in Figure 10 [58]. The performed heat treatment had a positive effect on a change in the microstructure of the investigated alloy, causing partial spheroidisation and coagulation of eutectic silicon precipitations in comparison with the microstructure of the alloy without the heat treatment.

On the side, the mechanical properties of 65Be-30.75Al-2Si-2Ag-0.04Sr-0.25 (Cu, Ni and Co) have really been raised after heat treated (solutionized, quenched, and aged), in which the transition elements Cu, Ni and Co were considered as potential Be phase strengtheners [31].

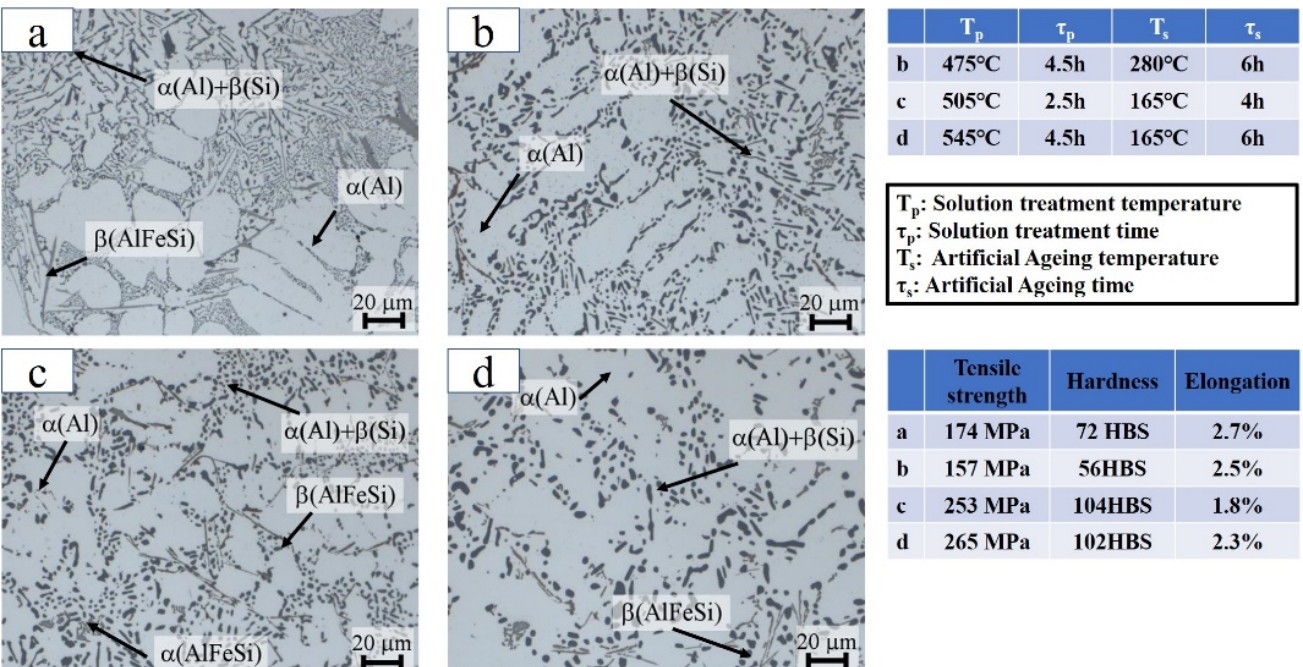

**Figure 10.** EN AC-AlSi11(Fe) alloy microstructure in the (**a**) initial state and after T6 heat treatment (**b**–**d**) [58].

The mechanical properties of 62Be-37.6Al-0.4Sc can also be improved by heat treatment when Al$_3$Sc phases in the as-cast state dissolved completely in the matrices. According to a previous study, there are numerous primary Al$_3$Sc particles with an average particle size of ~80 nm in the Al matrix of an as-cast Be-Al-0.4Sc alloy [59]. During the homogenizing process at 620 °C [59], these primary Al$_3$Sc phases in the as-cast state dissolved completely in the matrices after 2 h, resulting in an initial decrease in electrical conductivity and an increase in hardness, which was ascribed to the solid-solution strengthening [60] before this duration. The density of dislocations, which acted as heterogeneous nucleation sites for the Al$_3$Sc precipitates as the homogenizing time increased [61].

The mechanical property of Be-Al alloy can also be improved by hot isostatic pressing; it was found that the density of Be-Al alloy increased from 2.05 g/cm$^3$ to 2.14 g/cm$^3$, close to the theoretical density of alloy 2.17 g/cm$^3$ after hot isostatic pressing by Northwest Rare Metal Materials Research Institute Ningxia Co., Ltd. The temperature is selected as 550–580 °C in order to prevent Al from melting out, and the pressure is 60–80 MPa for 2 h.

## 5. The Defects Repair of Be-Al Alloy Casting

Cast Be-Al alloy is prone to defects due to its solidification characteristics. The defects of Be-Al alloy casting are more than those of conventional Al alloy. Welding is a common method to repair casting defects and electron beam welding (EBW) is often used to repair the casting alloys. The EBW has more concentrated energy and less thermal deformation; therefore it can be used to weld the Be-Al alloy in order to obtain the better mechanical properties, reported by Materion Inc. in the United States. The EBW repaired sections (Figure 11) have been tested utilizing tensile specimens that are equivalent to the weld repair section, with results that are equal to the tensile properties of the base material [62].

It can be combined the topological optimization of 3D printing with welding technology to promote the intelligent development of Be-Al alloy welding. Many research studies have been carried out on this topic. The Be-Al alloy was processed using laser remelting (LR) [2,63], and the microstructure was investigated using a 3D reconstruction technique. The defect repair of Be-Al alloy is a feasible method by laser 3D additive manufacturing [64].

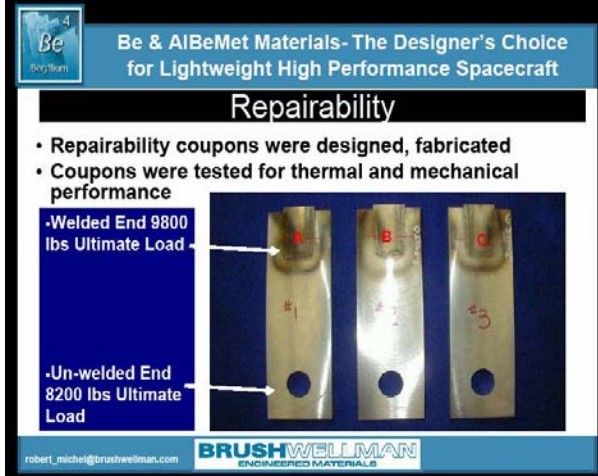
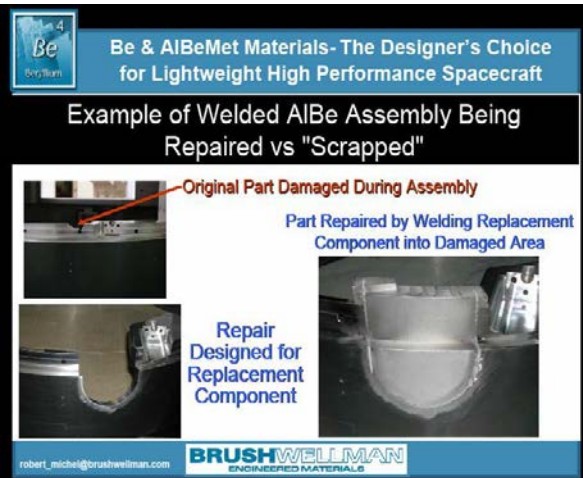

**Figure 11.** The EBW repaired sections of Be-Al alloy [62] Reprinted/adapted with permission from Ref. [62] Copyright 2008, SPIE.

## 6. The Development of Be-Al Alloy Casting in China

The research of Be-Al alloy in the United States began in the 1960s. The application of Be-Al alloy developed rapidly in the 1990s. Nowadays, the alloy has been applied to advanced weapons and equipment on a large scale. The Be-Al alloy was fabricated by investment casting process in 2005 by Northwest Rare Metal Materials Research Institute Ningxia Co., Ltd. The quality of Be-Al alloy casting has basically met the needs of domestic customers and solved the urgent need of Be-Al alloy during more than ten years of development. The investigation adding elements were Ag, Co, Ge, and Ni; its research mainly focuses on product performance improvement and application promotion, research on the microstructure and phase interface of materials is not enough.

In recent years, the Institute of Materials Research of China Academy of Engineering Physics has started research on Be-Al alloy and innovatively proposed research on Be-Al systems containing Sc. There are numerous theoretical bases for the research, which can provide a reference for promoting the theoretical development of Be-Al alloy in China. The disadvantage is that the performance of the alloy needs to be further enhanced to meet the application needs of customers.

However, the technology of Be-Al alloy casting in China needs to be further improved. The standard of Be-Al alloy casting is not perfect. As the cost of Be-Al alloy casting is still high, its application and market promotion are difficult. In addition, lack of understanding of the toxicity of Be toxicity for the customers.

In the next few years, the following advisers could be drawn.

- It should further accelerate the improvement of Be-Al alloy casting technology and equipment and improve casting product quality and yield in order to reduce the price of Be-Al alloy products. At the same time, standards and systems for Be-Al alloy products based on the casting Al alloy model should be established as soon as possible. More importantly, the construction of the Be-Al alloy product system must be accelerated according to the different application requirements of customers. For example, the development of a different Be content in alloy and the different processing technology of alloy series; it is also necessary to strengthen the collaboration between enterprises and universities, promote the basic theoretical and applied research of Be-Al alloy, establish the genetic database of Be-Al alloy, and provide technical support for the casting simulation and new product development of Be-Al alloy.
- The construction of the standard of Be-Al alloy casting needs to be accelerated, especially the establishment of casting defects evaluation and usability evaluation system. The Be-Al alloy products and the standard manufacturing system in China require

the form combined with the characteristics of the user. As a result, the process of "production, study, research and application" of Be-Al alloy will be accelerated.

- The defects repairing technology of Be-Al alloy casting should be studied as soon as possible. Due to the characteristics of Be-Al alloy, it is feasible to repair casting defects with laser and electron beams. In the future research plan, it is important to study the thermodynamics and kinetics of laser and electron beam welding of Be-Al alloy and further develop the specific process. On the other hand, it is necessary to develop the application of additive manufacturing technology, semi-solid casting and low-pressure casting technology in the preparation of Be-Al alloy.

- Technical communication with customers should be strengthened in order to improve customers' objective understanding of Be-Al alloy and promote the rapid development of Be-Al alloy. Most importantly, the research on industrial preparation technology of the alloy should be paid attention to, and the application scale of the alloy in the civil field should be continuously enhanced.

## 7. Conclusions

(1) It is necessary to add Ag, Co, Ge, Ni, Sc, Si and Cu elements to improve its casting properties and mechanical properties, mainly by strengthening the aluminum phase to improve the alloy strength. Be-Al alloy (Ag/Co/Ge) has been applied widely, such as AlBeCast 920® and BerylCast363®. Whereas, the research of Be-Al alloy added elements is less compared with aluminum alloy series, and theoretical research should be carried out in-depth. On the other hand, the phase interface of Be-Al alloy as a metal matrix composite requires further study.

(2) As with cast Al, one of the effective ways to improve the strength of Be-Al alloy casting is strengthening the Al phase by heat treatment. Meanwhile, the defects of Be-Al alloy can be repaired by the welding method, and the best way to do this is through electron beam welding. Furthermore, it can be combined the topological optimization of 3D printing with welding technology to promote the intelligent development of Be-Al alloy welding in future research.

(3) In the end, the development of Be-Al alloy casting in China is deeply analyzed, giving a significant expansion and increase in descriptive and comparative parts. Firstly, it is necessary to strengthen the basic theory and application of Be-Al alloy. Secondly, while promoting the application of Be alloy in the military field, it is essential to actively develop the application in the civil sector such as computer, electronics and instrument industries. Thirdly, new casting techniques and methods should be researched in order to reduce the manufacturing cost of Be-Al alloy.

**Author Contributions:** J.L.: Funding acquisition, Supervision, Resources, Validation, Writing—review & editing. Y.X.: Funding acquisition, Writing—original draft. Y.Y. (Yiqun Yang): Resource, Validation. Z.L.: Resources, Validation, D.W.: Funding acquisition, Supervision, Project administration, Y.Y. (Yajun Yin): Resource, Validation. All authors have read and agreed to the published version of the manuscript.

**Funding:** This work is supported by the National Key Research and Development Program of China (2021YFC2902304). China Nonferrous Metal Mining (Group) Co., Ltd. science and technology planning projects (2018KJJH02, 2020KJJH04), The central government directs special projects for the development of local science and technology (2020) and Ningxia Hui Autonomous Region Industrial Innovation List Unwrapping Project (20200108).

**Institutional Review Board Statement:** Not applicable.

**Informed Consent Statement:** Not applicable.

**Data Availability Statement:** The data presented in this study are available on request from the corresponding author. The data are not publicly available due to privacy.

**Conflicts of Interest:** The authors declare no conflict of interest.

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
