# Peer review of "Research Progress of Low Density and High Stiffness of Be-Al Alloy Fabricated by Investment Casting"

_metals, doi:10.3390/met12081379_

Round 1

Reviewer 1 Report

1. Fig. 1 and Fig. 2 captions: What the composition of the alloys provided in the Fig. 1 and Fig. 2? Please provide it in the manuscript text.

2. Table 3: The columns in Table 3 do not coincide with each other and because of that the table is difficult to read.

3. Page 5, line 150: What means the Be element here, it is also a phase - Be solid solution, like Al solid solution.

4. Figure 4 caption: What the composition of alloy for that EDS map is provided? Please show its composition in the figure caption.

5. Page 7, line 199: What "crust" means here?

6. Page 10, line 299-307: In that paragraphs, information about Al alloy is provided. Please explain how this connected with the Be alloys?

Author Response

We are very grateful for your expert comments, which can considerably enhance the quality of our manuscript. As the reviewer’s good instructions, we have corrected all the required revisions point by point as follows, which we hope meet with your approval.

Point 1: 1. Fig. 1 and Fig. 2 captions: What the composition of the alloys provided in the Fig. 1 and Fig. 2 ? Please provide it in the manuscript text.

Response 1: We have provided the composition of the alloys in Fig. 1(b) and Fig. 2. Thank you.

Point 2: Table 3: The columns in Table 3 do not coincide with each other and because of that the table is difficult to read.

Response 2: We have revised the columns form Table 3 in order to understand more clearly. Thank you.

Point 3: Page 5, line 150: What means the Be element here, it is also a phase - Be solid solution, like Al solid solution.

Response 3: We change Be element Page 5, line 150 to Be phase. Thank you

Point 4: Figure 4 caption: What the composition of alloy for that EDS map is provided? Please show its composition in the figure caption.

Response 4: The Be-Al alloy is 62wt.% Be-33.25 wt.% Al(3wt.%Ag/1 wt.%Co/0.75 wt.%Ge) in Figure 4. The compositions are shown in line 170. Thank you.

Point 5: Page 7, line 199: What "crust" means here?

Response 5: “crust” means “continental crust” in order to indicate that the reserves of Sc element on the earth are higher than those of Ag, Co, Pb and Sn. We eventually removed the expression.  

Point 6: Page 10, line 299-307: In that paragraphs, information about Al alloy is provided. Please explain how this connected with the Be alloys?

Response 6: Be-Al alloy should be defined as a composite material, in which the discontinuous granular Be phase reinforces the matrix of continuous Al phase. Heat treatment caused grain growth and the insoluble phases Si of Be-Al alloy aggregated and coarsened, resulting in an enhancement of tensile strength, similar to Al-Si alloy. Therefore, we provide information about Al alloy in 299-307.

Thank you once again!

Reviewer 2 Report

I have no detailed comments.

The manuscript provides a useful description of research progress of low density and high stiffness of Be-Al alloy fabricated by investment casting. The manuscript is a review and is primarily based on the results of the work of other researchers. The work is interesting and may be of value to researchers working with beryllium alloys.

The reviewer thinks it is a valuable work and deserves for publishing in MDPI Metals.

Author Response

Thanks for the reviewer's affirmation of the article. The paper is a review based on the results of the work of our and other researchers. We hope the paper is valuable to researchers working with beryllium alloys.

Thank you again very much!

Reviewer 3 Report

The paper “metals-1844816” related to casting was reviewed. Please follow the comments carefully and resubmit your paper for the next consideration and reviewing process.

  1. What is the main novelty of the paper? Please highlight it.
  2. Improve the abstract by adding short quantitative results to the abstract.
  3. Please double-check the values in Table 1.
  4. How authors selected the process parameters?
  5. Please explain Figure 1? The current explanation of this figure is not sufficient enough.
  6. Casting has many usages in different industries. To improve the contribution of the paper, add a short statement in the introduction and compare the casting with Additive Manufacturing and add the following papers.

·        The effect of absorption ratio on meltpool features in laser-based powder bed fusion of IN718

·        Effects of fused filament fabrication parameters on the manufacturing of 316L stainless-steel components: geometric and mechanical properties

·        Study on the impact behaviour of arch micro-strut (ARCH) lattice structure by selective laser melting (SLM)

·        Optimization of LB-PBF process parameters to achieve best relative density and surface roughness for Ti6Al4V samples: using NSGA-II algorithm

Author Response

We are very grateful for your valuable comment on our manuscript! As the Reviewer's good instruction, we have corrected the required revision in the manuscript and improved the readability of the text, which we hope meet with your approval.

Point 1: What is the main novelty of the paper? Please highlight it.

Response 1: The main novelty of the paper is that we introduce the Be-Al alloy, demonstrated in the following four aspects.

  • Some elements are often added to improve the mechanical properties and decrease casting defects of Be-Al alloy.
  • The strength of Be-Al alloy casting can be improved by heat treatment.
  • Welding is a common method to repair casting defects and electron beam welding (EBW) is often used to repair the castings.
  • The development of Be-Al alloy casting in China.

We highlight them in green colour in the paper.

Point 2: Improve the abstract by adding short quantitative results to the abstract.

Response 2: We have made changes to the abstract as requested.

Point 3: Please double-check the values in Table 1.

Response 3: We have checked Table 1 again in detail, and unscientific data were revised and deleted.

Point 4: How authors selected the process parameters?

Response 4: We have introduced the process parameters selected in the paper in detail in line 145-154.

Point 5: Please explain Figure 1? The current explanation of this figure is not sufficient enough.

Response 5: Thank you very much for your recommendation! We added some notes to illustrate Figure 1 in more detail.

Point 6: Casting has many usages in different industries. To improve the contribution of the paper, add a short statement in the introduction and compare the casting with Additive Manufacturing and add the following papers.

  • The effect of absorption ratio on meltpool features in laser-based powder bed fusion of IN718

  • Effects of fused filament fabrication parameters on the manufacturing of 316L stainless-steel components: geometric and mechanical properties

  • Study on the impact behaviour of arch micro-strut (ARCH) lattice structure by selective laser melting (SLM)

  • Optimization of LB-PBF process parameters to achieve best relative density and surface roughness for Ti6Al4V samples: using NSGA-II algorithm

Response 6: We have read the literature recommended by the reviewer in detail, added a short statement in the introduction, add the four papers from the reviewer and compare the casting with Additive Manufacturing(AM). What is more, the patterns used in investment casting of Be-Al alloy can be prepared by additive manufacturing technology(SLS).

Round 2

Reviewer 3 Report

The paper is ready to publish.

This manuscript is a resubmission of an earlier submission. The following is a list of the peer review reports and author responses from that submission.

Round 1

Reviewer 1 Report

Review of the manuscript entitled “Research on Low Density and High Stiffness of Be-Al Alloy Fabricated by Investment Casting”. The topic is investment casting of Be-Al alloy.
The manuscript IS NOT a research article, as presented by the authors. Structure and contents are not in line with the description.
It is not declared the main goal of the manuscript. Discussion is not present.
Presented results do not come from experiments of the authors but are just taken from literature. However it can’t be considered a review article because extremely short and limited. Ref. from 37 to 66 are not called in the main text. References (all) are not in compliance with the journal requirements.
I cannot see the main results (and novelty) of the paper. I cannot clearly see the problem that the authors intend to solve.
In my opinion the manuscript doesn’t meet the quality level of the Journal and must be rejected. 

Author Response

Point 1: The topic is investment casting of Be-Al alloy. The manuscript IS NOT a research article, as presented by the authors. Structure and contents are not in line with the description.

Response 1: We thank the journal reviewer for constructive comments and suggestions. This paper is a review of the research progress of investment casting Be-Al alloy. The structure and content of the article have been readjusted in the revised manuscript.

Point 2: It is not declared the main goal of the manuscript. Discussion is not present.

Response 2: The main goal of the manuscript is investment casting Be-Al alloy, including the structure of the alloy, strengthening mechanism of adding elements, heat treatment and welding, in the end, the development of Be-Al alloy casting in China are also presented in detail.

Point 3: Presented results do not come from experiments of the authors but are just taken from literature.

Response 3: Some part of presented results come from our previous experiments, meanwhile, we also cited other research results to complete this paper.

Point 4: Presented results do not come from experiments of the authors but are just taken from literature. However it can’t be considered a review article because extremely short and limited. Ref. from 37 to 66 are not called in the main text. References (all) are not in compliance with the journal requirements.

Response 4: This new manuscript has been fully revised, reorganized, and updated to make its content even more accessible. More importantly, we put the reference numbers in the right order as requested carefully, and there are 56 references in the revised version in accordance with the journal requirements.

Reviewer 2 Report

The article Research on Low Density and High Stiffness of Be-Al Alloy Fabricated by Investment Casting is devoted to the study of the properties of Be-Al alloy, which has a high potential for application in the aerospace industry. Undoubtedly, the results presented by the authors are of high scientific novelty and practical significance, and are also promising for practical research. In general, the presented results of the study can be accepted for publication after the authors provide answers to all the questions raised by the reviewer during the reading of the article.

1. In the abstract, the authors need to more clearly state the purpose and relevance of this work.
2. The authors should explain the data presented on the SEM image on how aluminum and beryllium grains interact, and also compare them with similar structures of titanium beryllide, which has a similar structure.
3. Also, the authors should provide explanations about the choice of dopants presented in Table 1.
4. The abstract provides data on strength properties, however, from the data presented in the article, there is no exact idea of ​​their change depending on the concentration of dopants or temperature heating.
5. Section 6. The development of Be-Al alloy casting in China requires a significant expansion and increase in the descriptive and comparative part.
6. The conclusion requires additions.

Reviewer 3 Report

General review: The authors describe Be-Al alloy having a low density and high elastic modulus, and it will become a preferred material for lightweight aerospace products in the future. Investment casting technology can be used to prepare the products with thin-walled complex structures for aerospace. This paper introduces the characteristics of Be-Al alloy, the influence mechanism of adding elements on casting process and properties, and the heat treatment technology. The problems and counter-measures in the research of Be-Al alloy in China are analyzed. However, before it can be published, I have some questions about this article and some suggestions:

Minor revision:

  1. Even though I understood the effectiveness of the Be-Al alloy, you should mention about the toxicity of the Be in the experimental or results and discussion in brief.
  2. For the section of The heat treatment of casting Be-Al alloy, is there any difference between the heat treatment of T5 or T6 for Al-Si alloy and that of the cast Be-Al alloy?
  3. For the section of 5. The defect repair of casting Be-Al alloy, the concept seems to be similar to the topological optimization of 3D printing. You should read and cite the paper, “Additive manufacturing of a shift block via laser powder bed fusion: the simultaneous utilisation of optimised topology and a lattice structure”.
  4. You better replace from the old references of more than 10 years to the more recent ones.

Round 2

Reviewer 1 Report

The manuscript has been significantly improved and it can be accepted for publication in the present form.

Reviewer 2 Report

The authors answered all questions, the article can be accepted for publication.